

# Sleep and memory complaints in long COVID: an insight into clustered psychological phenotypes

Ricardo Titze-de-Almeida[1,2], Pedro Henrique Araújo Lacerda[1], Edson Pereira de Oliveira[1], Mariah Eduarda Ferreira de Oliveira[1], Yngrid Sallaberry Silva Vianna[1], Amanda Machado Costa[1], Eloísa Pereira dos Santos[1], Louise Marie Coelho Guérard[1], Matheus Augusto de Miranda Ferreira[1], Isabel Cristina Rodrigues dos Santos[1], Jéssica Daniele da Silva Gonçalves[1], Gabriel Ginani Ferreira[1,2], Simoneide Souza Titze-de-Almeida[1,2], Pedro Renato de Paula Brandão[1,3], Helena Eri Shimizu[4], Andrezza Paula Brito Silva[1], Raimundo Nonato Delgado-Rodrigues[1] and Research Center for Major Themes–COVID-19 group

[1] Central Institute of Sciences, Research Center for Major Themes, University of Brasília, Brasília, DF, Brazil
[2] University of Brasília/FAV, Central Institute of Sciences, Technology for Gene Therapy Laboratory, Brasília, DF, Brazil
[3] Sírio-Libanês Hospital, Brasília, Brazil., Brasília, DF, Brazil
[4] Department of Collective Health, Research Center for Major Themes, University of Brasília, Brasília, DF, Brazil

Corresponding author
Ricardo Titze-de-Almeida,
ricardotitze.unb@gmail.com

## ABSTRACT

This study evaluated clinical features of individuals with long COVID (5–8 months after diagnosis) who reported sleep and memory problems (62 cases) compared to those without (52 controls). Both groups had a similar mean age (41 *vs.* 39 years). Around 86% of the participants were non-hospitalized at the time of infection, and none of them were vaccinated at that point. Subsequently, both cases and controls received the vaccine; however, the vaccination rates differed significantly between the groups (30.7% *vs.* 51.0%). Cases and controls had similar rates of symptoms at acute COVID phase. However, cases were more likely to experience coryza, dyspnea, headache, and nausea/vomiting during long COVID. Regarding new-onset symptoms in long COVID, 12.9% of cases had dyspnea, and 14.5% experienced nausea/vomiting, whereas in the control group there were only 1.9% and 0.0%, respectively. Cases also had a significantly higher prevalence of persistent headache (22.6% *vs.* 7.7%), and dyspnea (12.9% *vs.* 0.0). In addition, cases also showed an increased rate of mental health complaints: disability in daily activities (45.2% *vs.* 9.6%; $P < 0.001$); concentration/sustained attention difficulties (74.2% *vs.* 9.6%; $P < 0.001$); anxiety–Generalized Anxiety Disorder 2-item scale (GAD-2) $\geq 3$ (66.1% *vs.* 34.6%; $P = 0.0013$); and "post-COVID sadness" (82.3% *vs.* 40.4%; $P < 0.001$). We observed a significant correlation between sadness and anxiety in cases, which was not observed in controls ($P=0.0212$; Spearman correlation test). Furthermore, the frequency of concomitant sadness and anxiety was markedly higher in cases compared to controls (59.7% *vs.* 19.2%) ($P < 0.0001$; Mann-Whitney test). These findings highlight a noteworthy association between sadness and anxiety specifically in cases. In conclusion, our data identified concurrent psychological phenotypes in individuals experiencing sleep and memory disturbances during long COVID. This strengthens the existing evidence that SARS-CoV-2 causes widespread brain pathology with interconnected phenotypic

clusters. This finding highlights the need for comprehensive medical attention to address these complex issues, as well as major investments in testing strategies capable of preventing the development of long COVID sequelae, such as vaccination.

## INTRODUCTION

In most COVID-19 cases, symptoms typically resolve within several weeks following diagnosis. However, a significant subset of patients experience long COVID, a condition defined by the emergence of new or persistent symptoms that occur at least four weeks after initial diagnosis and persist for a minimum of two months, in the absence of any other identifiable cause (*O'Mahoney et al., 2023*; *Soriano et al., 2022*). Long COVID manifests as a multi-organ disease with a wide range of symptoms, encompassing dyspnea, fatigue, myalgia, muscle weakness, and neurological complaints such as headache, anosmia, and cognitive difficulties (*Ghosn et al., 2021*; *Huang et al., 2021*; *Kubota, Kuroda & Sone, 2023*; *Lund et al., 2021*; *Michelen et al., 2021*; *Nalbandian et al., 2021*), as previously reported by our group (*Titze-de-Almeida et al., 2022*).

Long COVID may arise from a complex network of underlying mechanisms (*Crook et al., 2021*; *Davis et al., 2023*). While SARS-CoV-2 particles typically clear from the body within a few weeks after infection and may not directly contribute to long COVID development, genome RNA fragments and spike proteins persist for months and may elicit delayed pathological processes (*Griffin, 2022*; *Marshall, 2021b*; *Swank et al., 2022*).

Patients with neurological sequelae show increased levels of biomarkers of neuronal loss and cytokines that trigger persistent immune responses, as occurs in auto-immune diseases (*Sun et al., 2021*). Furthermore, the infection of astrocytes and changes in brain vasculature may also contribute to long COVID neuropathology (*Davis et al., 2023*; *Marshall, 2021a*).

Studies conducted during this pandemic have demonstrated an increase in the rates of psychiatric and neuropsychiatric disorders that affect cognitive, affective, behavioral, and perceptual functions (*Rogers et al., 2020*). Initially, studies involving sleep during the onset of the COVID-19 pandemic seemed to suggest that insomnia or poor sleep quality were primarily due to psychosocial factors, such as confinement and anxiety related to the loss of economic resources (*Efstathiou et al., 2022*; *Robillard et al., 2021*), or by the disease itself, as well as fear/stress of infection and the prospect of financial damages from a long hospitalization (especially at the time of data collection of this group of patients, when effective means of COVID-19 prevention were not yet available) stand out. On the other hand, other studies have also found evidence of a brain inflammatory state exacerbated by sleep deprivation/fragmentation, a phenomenon demonstrated in obstructive sleep apnea syndrome for several years (*Gabryelska et al., 2020*; *Semyachkina-Glushkovskaya et al., 2021*).

Mental health outcomes in long haulers represent a major issue in public health due to their impact on individuals' quality of life and work productivity (*Marshall, 2021b*;

*Mizrahi et al., 2023*; *Nalbandian et al., 2021*; *Soriano et al., 2022*). The most frequent complaints include insomnia, anxiety, and depressive and post-traumatic stress symptoms (*The Writing Committee for the COMEBAC Study Group et al., 2021*; *Huang et al., 2021*; *Mei et al., 2021*; *Naidu et al., 2021*; *Orru et al., 2021*; *Taquet et al., 2021*). Indeed, mental health complaints in long COVID commonly arise as clusters of associated phenotypes, including deficits in cognitive functions and headache (*Evans et al., 2021*; *Kenny et al., 2022*).

The current research deepens our previous work on clinical manifestations of acute and long COVID (*Titze-de-Almeida et al., 2022*). One of the key findings of this research was the significant association between memory problems, sleep problems, and "post-COVID sadness". Number wise, 68.1% of the individuals relating to memory problems also report sleep complaints ($P = 0.0003$; adjusted OR 3.206, 95% CI [1.723–6.030]). In the same way, 69.1% of the individuals with memory problems report "post-COVID sadness" in long COVID ($P < 0.0001$; adjusted OR 3.981, 95% CI [2.068–7.815]).

In the same study, we also found that sleep disturbance was the most prevalent mental phenotype, affecting 46% of subjects. Additionally, memory complaints were highly prevalent, occurring in 40% of individuals in that cohort. Moreover, we observed a significant increase in the occurrence of sleep disturbance and depression (but not anxiety) in a subgroup of individuals who reported memory problems. These findings from our previous data led us to propose a hypothesis that different mental health complaints in long COVID may occur in the same individuals as a cluster of symptoms, indicative of a widespread brain pathology caused by SARS-CoV-2. While the above-mentioned studies have reported clusters of post-COVID symptoms, they have not specifically focused on memory and sleep (*Evans et al., 2021*; *Kenny et al., 2022*).

Previous studies have also identified higher rates of sleep and memory problems during long COVID, highlighting the need for further investigations into these debilitating health issues caused by pandemics. *Zhao et al. (2022)* found impaired vigilance and episodic memory at 6 months after COVID-19 infection in individuals with low symptom burden (*Zhao et al., 2022*). Their study included 36 cases and 44 controls, similar to our study involving non-hospitalized patients without ICU or post-COVID care. *Stavem et al. (2022)* also reported a decline in short-term memory, visuospatial processing, learning, and attention at 11 months post-COVID-19 infection in a cohort of non-hospitalized patients ($n = 234$) (*Stavem et al., 2022*). Interestingly, this study found no association between symptom severity and impairment, suggesting that the neurological burden due to COVID-19 might play a prominent role in mental consequences compared to the occurrence of physical symptoms.

In a comprehensive analysis of over 18,000 patients from 51 studies, sleep disturbance emerged as the most prevalent neuropsychiatric symptom, affecting approximately 27.4% of individuals (*Badenoch et al., 2022*). Other common symptoms included fatigue (24.4%), cognitive impairment (20.2%), anxiety (19.1%), and post-traumatic stress (15.7%). Surprisingly, the severity or duration of the initial COVID-19 infection did not appear to be significantly associated with the persistence of these symptoms. Similarly, *Fernandez-de-Las-Penas et al. (2021)* found that poor sleep quality affected 34.5% of individuals in Spain

diagnosed with COVID-19, followed by depressive symptoms and anxiety at 7 months post-diagnosis in 2021.

A nationwide study conducted in Denmark found that 10.9% of COVID-19-positive individuals experienced sleep problems. Sleep problems were among the top three most common issues in COVID-19-positive individuals, along with fatigue/exhaustion and dysgeusia (*Sorensen et al., 2022*). Additionally, *Brown et al. (2022)* conducted a study involving mostly non-hospitalized individuals with a mean age of 46 years and found that perceived sleep concerns, rather than posttraumatic stress disorder symptoms or anxiety symptoms, could predict self-reported memory disturbances (*Brown et al., 2022*).

Considering the aforementioned relevance of sleep and memory problems during long COVID, this study aimed to evaluate the clinical characteristics of long COVID patients (*i.e.,* individuals experiencing symptoms 5–8 months after diagnosis) who reported sleep and memory complaints, in comparison to a control group without such issues. The study assessed the prevalence of daily activity impairment, difficulties in concentrating/sustaining attention, anxiety, and "post-COVID sadness" in both groups. Furthermore, the study investigated whether individuals with sleep and memory complaints exhibited co-occurring phenotypes of anxiety and "post-COVID sadness", which potentially indicate a widespread brain pathology caused by SARS-CoV-2.

## MATERIALS & METHODS

### Sample

The present study included a subset of RT-qPCR–confirmed cases of COVID-19 that matched the current criteria for long COVID, which were part of a cohort from 'Hospital Regional de Santa Maria' (HRSM) and 'Hospital de Base do Distrito Federal' (HBDF) previously evaluated by this group (*Titze-de-Almeida et al., 2022*).

Patient data from individuals over 18 years old were collected in two periods. The first period was from September 2020 to December 2020, during which COVID-19 infection was diagnosed. Data was collected during the first two weeks after a positive SARS-CoV-2 RT-qPCR result, referred to as the acute disease phase. The second period for data collection was in May 2021, 5–8 months after the RT-qPCR test positivity, named the long COVID phase.

All individuals were non-vaccinated at the time of their SARS-CoV-2 infection from September 2020 to December 2020. Subsequently, some individuals received one or two vaccine doses between January 2021 and May 2021. Additional socio-demographics and clinical information is presented in the results section.

The study adhered to ethical guidelines and received approval from the Ethics Committee at the Institute of Strategic Health Management of the Federal District (IGESDF) under the Brazilian Platform (Plataforma Brasil). All participants provided signed and informed consent, and the study was assigned the Certificate of Presentation of Ethical Appreciation (Certificado de Apresentação de Apreciação Ética–CAAE) number 36147920.1.0000.8153.

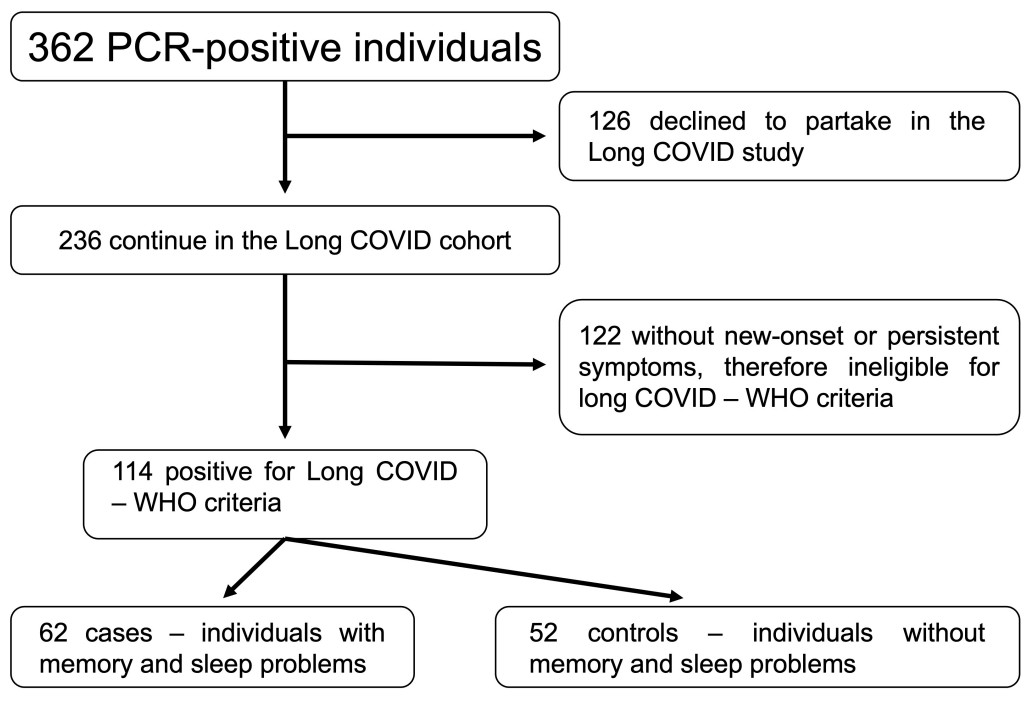

**Figure 1  Flowchart of the study process.** Abbreviations: PCR, polymerase chain reaction; COVID-19, coronavirus disease 2019; WHO, World Health Organization.

## Long COVID criteria

Long COVID can be defined as a syndrome of new or persistent symptoms manifested a month after the initial diagnosis of COVID-19, which are not related to other conditions and last for approximately 2 months (*Soriano et al., 2022*). In other words, those symptoms had to appear or even worsen after the acute phase of the infection have passed (*Gasnier et al., 2022*). The individuals that attend this condition were included in the study.

## Cases and controls definition

Long COVID individuals of this study were organized in two cohorts concerning their sleep and memory phenotypes. Cases were individuals who reported sleep and memory complaints, whereas controls were those without both phenotypes.

## Flowchart

As shown in the flowchart presented in Fig. 1, our first cohort comprised 362 individuals with positive RT-qPCR tests for COVID-19 at the beginning, and some of them refused to participate in the study ($n = 126$). Besides, some were non-eligible according to OMS criteria for long COVID-19 (*Soriano et al., 2022*) ($n = 122$). We finally set our study cohort of 114 individuals, organized in cases ($n = 62$) and controls ($n = 52$).

## Data survey

A data survey was obtained *via* phone calls, which gathered both demographic and clinical information for the acute and chronic phases of COVID-19. Structured questionnaires were

applied in those calls to assess typical COVID-19 symptoms and long COVID complaints, with emphasis on sleep complaints, the post-COVID onset of "post-COVID sadness", anxiety, memory, and other phenotypes, as previously described (*Titze-de-Almeida et al., 2022*). All data generated was managed through a SQL (Structured Query Language) relational database designed for the project. We employed a 'Team Desk' platform which uses virtual cloud processing and cryptographed backups of all information.

## Statistical analysis

Categorical variables were represented as the count and percentage of occurrences. The reported symptoms from acute and long COVID phases were grouped into four categories: 1–2, 3–5, 6–8, and 9–12. Symptoms present during both phases were classified as 'persistent', while symptoms that appeared only in the long COVID phase (5–8 months after diagnosis) were labeled as 'new-onset symptoms'. Statistical analyses were performed using Prism 9 software (Version 9.1.2). A significance level of $P < 0.05$ was used to determine statistical significance.

## RESULTS

This section presents an overview of the demographics and clinical characteristics of the study population. The study results are divided into two parts based on the research goals. Part 1 aims to describe the frequency of typical COVID-19 symptoms in both cases and controls, while Part 2 focuses on cognitive and psychiatric conditions in these groups.

Our cohort consisted of 114 individuals who developed long COVID, divided into two groups: cases ($n = 62$), who reported sleep and memory problems, and controls ($n = 52$) without such complaints. As shown in Table 1, the mean age of cases and controls was 41.5 and 38.9, respectively, with a higher proportion of females in both groups (75.8% and 65.4%, respectively). Positive RT-qPCR diagnostics were distributed across the months of September to December 2020, and the rates were not statistically different between cases and controls. Although cases and controls were not vaccinated when the infection occurred, a lower proportion of cases received vaccines compared to controls after the infection (30.7% *vs.* 51.9%; *$p < 0.05$, Fisher's exact test). The rates of frequent comorbidities, such as overweight/obesity, hypertension, and diabetes, did not show significant differences between the cases and controls. In both groups, the majority of individuals were not hospitalized, with percentages of 87.1% and 86.5% for cases and controls, respectively. Additionally, only a subset of participants (11% in both groups) required oxygen supplementation.

### Part 1–COVID-19 symptoms in cases and controls

We first evaluated the frequency of typical COVID-19 symptoms in acute and long COVID (5 to 8 months after diagnosis) in cases, individuals with sleep and memory problems, and controls, those without both complaints. As shown in Table 2, certain symptoms were prevalent in cases, such as myalgia, hyposmia and dysgeusia, as well as headache, present in more than 50% of cases in acute phase. The control group also reported myalgia, hyposmia, dysgeusia, and headache as the most common symptoms in acute phase. The

**Table 1  Demographic and clinical characteristics comparison between cases and controls.**

| Characteristic | Cases | Controls | P value |
|---|---|---|---|
| Age, mean years (SD) | 41.5 (11.6) | 38.9 (14.9) | 0.1128 |
| Woman, n (%) | 47 (75.8%) | 34 (65.4%) | 0.3000 |
| Period of COVID-19 RT-qPCR-confirmed diagnosis, n (%) | | | |
| September | 5 (8.1%) | 6 (11.5%) | 0.5448 |
| October | 22 (35.5%) | 22 (42.3%) | 0.5627 |
| November | 15 (24.2%) | 14 (26.9%) | 0.8300 |
| December | 20 (32.3%) | 10 (19.2%) | 0.1380 |
| Vaccination, n (%) | 19 (30.7%) | 27 (51.9%) | 0.0233** |
| Comorbidities, n (%) | | | |
| Overweight (pre-obesity) | 27 (45%) | 18 (39.1%) | 0.5594 |
| Obesity | 16 (26.7%) | 10 (21.7%) | 0.6514 |
| Essential hypertension | 9 (14.5%) | 10 (19.2%) | 0.6156 |
| Diabetes | 4 (6.5%) | 4 (7.7%) | >0.9999 |
| Chronic lung disorder (asthma, COPD) | 5 (8.1%) | 2 (3.9%) | 0.4515 |
| Chronic kidney disease | 1 (1.6%) | 0 (0%) | >0.9999 |
| Immunosuppression | 2 (3.2%) | 1 (1.9%) | >0.9999 |
| Heart disorder (coronary artery disease or valve disorder or heart failure) | 0 (0%) | 2 (3.9%) | 0.2059 |
| Neoplasia | 1 (1.6%) | 1 (1.9%) | >0.9999 |
| Solid-organ or bone marrow transplant | 0 (0%) | 1 (1.9%) | 0.4561 |
| No known diagnosis of chronic disorder | 30 (48.4%) | 29 (55.8%) | 0.4573 |
| Smoker, n (%) | 3 (4.8%) | 0 (0%) | 0.2492 |
| COVID-19 treatment scenario, n (%) | | | |
| Non-hospitalized patients | 54 (87.1%) | 45 (86.5%) | >0.9999 |
| Hospitalized patients (COVID-19 hospital ward) | 8 (12.9%) | 7 (13.5%) | >0.9999 |
| Critical care–intensive care unit (ICU) | 1 (1.61%) | 2 (3.9%) | 0.5911 |
| Oxygen supplementation | 7 (11.3%) | 6 (11.5%) | >0.9999 |
| Mechanical ventilation | 1 (1.61%) | 0 (0%) | >0.9999 |

**Notes.**
*Fisher's exact test, *$p < 0.05$.

frequency was slightly higher in cases when compared to controls, although not statistically significant in this acute phase. In long COVID, however, we found differences between cases and controls. First, cases had higher rates of headache, fatigue, myalgia, and dyspnea, while controls manifested mainly hyposmia, fatigue, dysgeusia, headache, and myalgia.

Cases were significantly more symptomatic than controls for coryza (14.5% *vs.* 0%; $P = 0.0037$), dyspnea (25.8% *vs.* 1.9%; $P = 0.00030$), headache (38.7% *vs.* 13.5%; $P = 0.0030$) and nausea/vomiting (21% *vs.* 1.9%; $P = 0.0029$) (Table 2, and Fig. 2).

We then explored the number of manifested symptoms, organized in the following groups for statistical comparisons: 1 to 2, 3 to 5, 6 to 8, and 9 to 12. Cases and controls showed no significant differences in the acute and long COVID phases (Fig. 3, upper panels).

**Table 2** Frequency of typical COVID-19 phenotypes.

| Phenotypes | Acute COVID | | | Long COVID | | |
|---|---|---|---|---|---|---|
| | Cases n (%) | Controls n (%) | P values[1] | Cases n (%) | Controls n (%) | P values[1] |
| Myalgia | 35 (56.5%) | 28 (53.8%) | 0.8508 | 18 (29%) | 7 (13.5%) | 0.0680 |
| Hyposmia/anosmia | 33 (53.2%) | 27 (51.9%) | >0.9999 | 11 (17.7%) | 12 (23.1%) | 0.4931 |
| Dysgeusia/ageusia | 33 (53.2%) | 25 (48.1%) | 0.7070 | 9 (14.5%) | 10 (19.2%) | 0.6156 |
| Fever | 24 (38.7%) | 20 (38.5%) | >0.9999 | 1 (1.6%) | 0 (0.0%) | >0.9999 |
| Fatigue | 26 (41.9%) | 19 (36.5%) | 0.5708 | 20 (32.3%) | 12 (23.1%) | 0.3027 |
| Dry cough | 22 (35.5%) | 20 (38.5%) | 0.8459 | 7 (11.3%) | 2 (3.8%) | 0.1777 |
| Coryza | 12 (19.4%) | 10 (19.%) | >0.9999 | 9 (14.5%)[**] | 0 (0.0%) | 0.0037 |
| Dyspnea | 19 (30.6%) | 12 (23.1%) | 0.4041 | 16 (25.8%)[***] | 1 (1.9%) | 0.0003 |
| Sore throat | 10 (16.1%) | 11 (21.2%) | 0.6285 | 6 (9.7%) | 2 (3.8%) | 0.2871 |
| Diarrhea | 9 (14.5%) | 9 (17.3%) | 0.7980 | 6 (9.7%) | 1 (1.9%) | 0.1236 |
| Headache | 32 (51.6%) | 23 (44.2%) | 0.4573 | 24 (38.7%)[**] | 7 (13.5%) | 0.0030 |
| Nausea/vomiting | 9 (14.5%) | 11 (21.2%) | 0.4595 | 13 (21%)[**] | 1 (1.9%) | 0.0029 |
| Loss of appetite | 10 (16.1%) | 9 (17.3%) | >0.9999 | 3 (4.8%) | 0 (0.0%) | 0.2492 |
| Abdominal pain | 5 (8.1%) | 4 (7.7%) | >0.9999 | 2 (3.2%) | 1 (1.9%) | >0.9999 |
| Expectoration | 5 (8.1%) | 0 (0.0%) | 0.0618 | 1 (1.6%) | 1 (1.9%) | >0.9999 |

**Notes.**
[**] $p < 0.01$.
[***] $p < 0.001$; Fisher's exact test.

Finally, we evaluated if the median of symptom numbers would be distinct between cases and controls (Fig. 3, bottom panels). In the same trend, the medians were four symptoms for cases and controls and showed no significant difference regarding the acute phase. In contrast, differences were significant during long COVID, and the medians were two symptoms for cases and one symptom for controls. Cases presented larger areas of violin plot, including the projection area above the median that was well established in cases and absent in controls, meaning that cases were more symptomatic in long COVID.

Our study also found differences in persistent and new-onset symptoms between cases and controls with long COVID. In cases, the most frequent persistent complaints were headaches and myalgia, affecting 22.6% of individuals, followed by dyspnea, fatigue, hyposmia, and dysgeusia, all with a percentage of 12.9% (Table 3 and Fig. 4, upper panel). In controls, the most common symptoms were hyposmia and dysgeusia, affecting 19.2% of individuals, followed by myalgia (13.5%) and fatigue (11.5%). Headache and dyspnea were persistent symptoms that occurred at a significantly higher rate in cases compared to controls (22.6% *vs.* 7.7%, $P = 0.0390$; and 12.9% *vs.* 0.0%, $P = 0.0075$, respectively).

Cases also showed higher rate of manifested new-onset symptoms, highlighting fatigue (19.4%), headache (16.1%), nausea/vomiting (14.5%), and dyspnea (12.9%). In controls, fatigue (11.5%) and headache (5.8%) were the most common symptoms, followed by hyposmia (3.8%). Two new-onset symptoms were significantly higher in cases compared to controls, dyspnea (12.9% *vs.* 1.9%; $P = 0.0380$) and nausea (14.5% *vs.* 0.0%; $P = 0.0037$). Finally, we found that the number of new-onset symptoms was significantly increased in cases, as shown in the violin plot in Fig. 4, bottom panel.

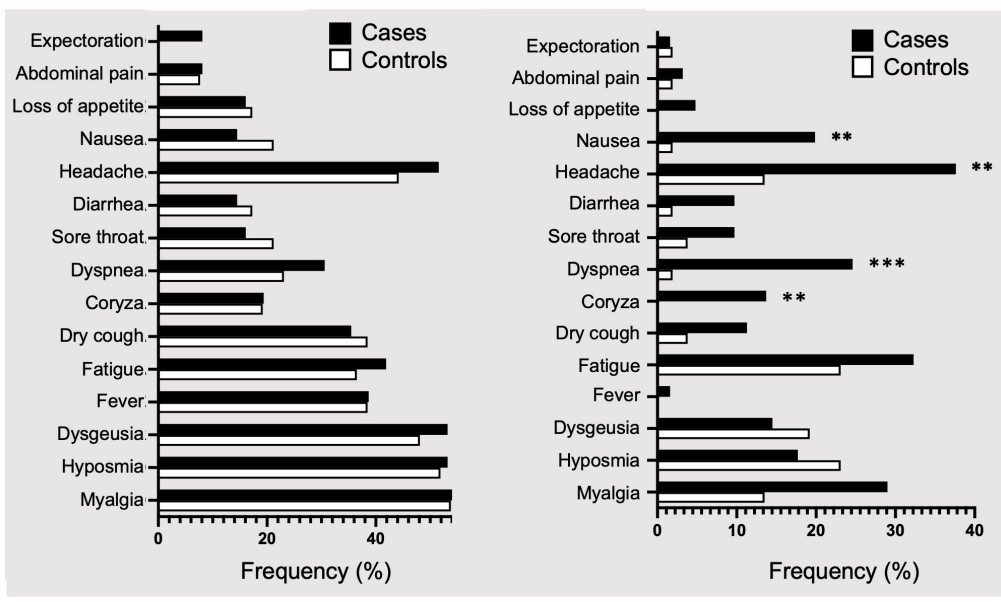

**Figure 2  Symptoms prevalence in cases and controls in acute and long COVID.** Percentages of COVID-19 symptoms in cases (black bars) and controls (white bars) during acute or long COVID, in patients that met the WHO long COVID criteria from two hospitals in mid-western Brazil. Abbreviation: COVID used to represent COVID-19 for simplicity, coronavirus disease 2019. Statistical analysis was performed using Fisher's exact test, with significance indicated by *** $p < 0.001$ or ** $p < 0.01$.

New-onset–and persistent symptoms were also divided into categories and analyzed. Concerning persistent symptoms, cases presented slightly increase in the total number of symptoms than controls (59.7% *vs.* 53.8%) and comprised slightly minor proportion of individuals with few symptoms, *e.g.*, up to two symptoms (46.8% *vs.* 53.8%) (Table 4). In the same direction, only cases had three or more persistent symptoms (12.9% *vs.* 0%; $P = 0.0075$). Regarding new-onset symptoms, cases were more symptomatic for all three categories: occurrence of new symptoms (45.2% *vs.* 26.9%, $P = 0.0527$), up to 2 new-onset symptoms (30.6% *vs.* 26.9%), and three or more new symptoms (14.5% *vs.* 0.0%, $P = 0.0037$) (Table 5). Indeed, only cases had three or more new-onset symptoms, with a significant difference for both (Fig. 5).

Finally, we analyzed the total number of symptoms shown by the individuals with acute COVID, long COVID, new-onset–and persistent symptoms (Table 6). Cases and controls showed no significant differences in the acute phase, which confirmed our previous data. The opposite occurred in long COVID, as cases contributed to 64.3% of the total symptoms and controls to only 35.7% ($P = 0.0134$). Cases were slightlty more symptomatic for persistent symptoms (56.6% *vs.* 43.4%; not statistically significant) and new-onset symptoms (76.4% *vs.* 23.6%; $P = 0.0108$).

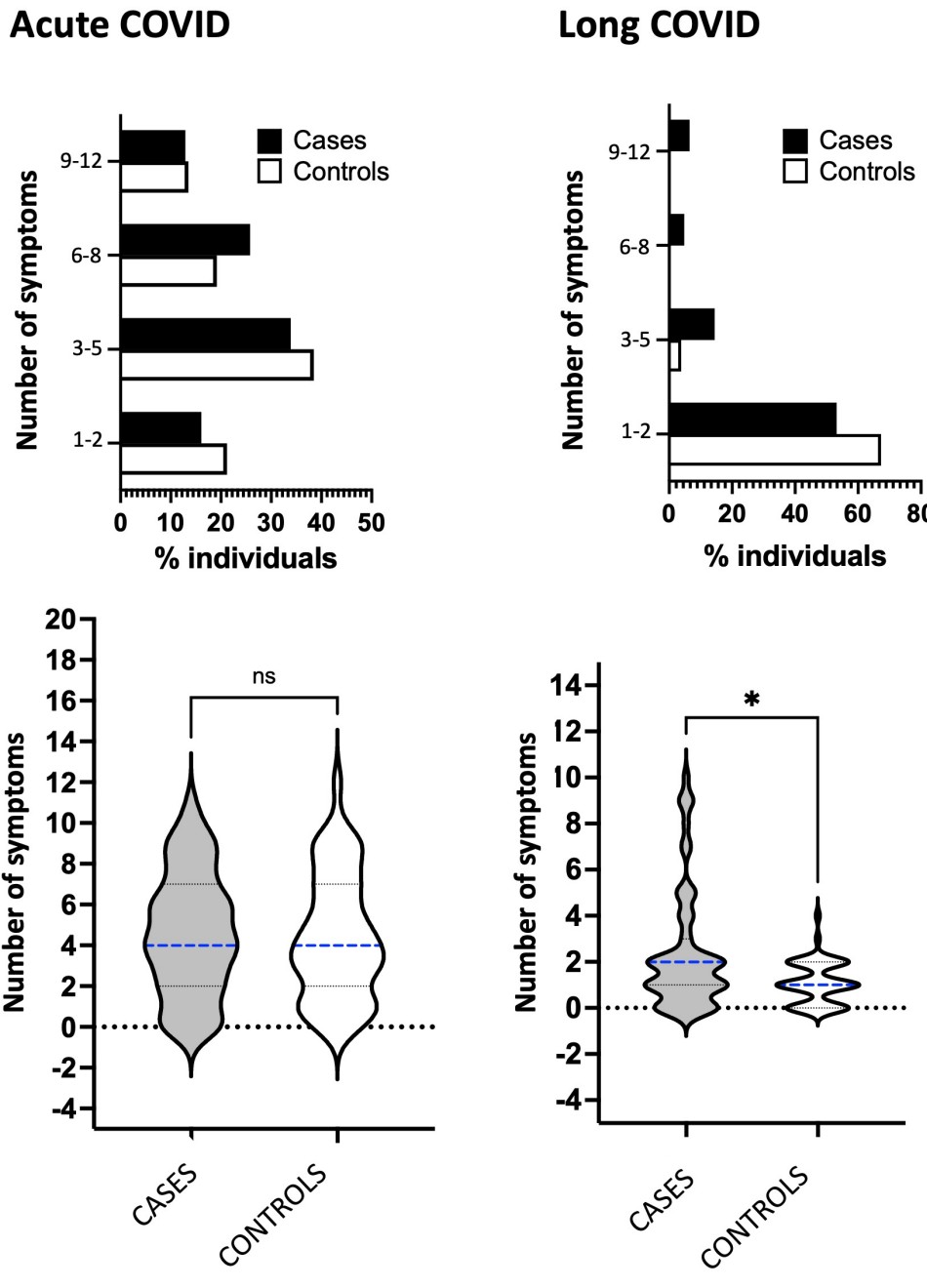

**Figure 3** **Analyses of the number of COVID-19 symptoms.** Percentage of symptoms categorized by number (1–2, 3–5, 6–8, and 9–12) for cases (represented by black bars) and controls represented by white bars) during the acute or long COVID phase. The bottom panel shows the total number of symptoms and median values for cases (represented by gray violin) and controls (represented by white violin) in acute or long COVID. The abbreviation "COVID" refers to coronavirus disease 2019. Statistical analysis was performed using Fisher's exact test, with significance indicated by $* p < 0.05$.

**Table 3  Persistent and new-onset COVID-19 symptoms.**

| *Phenotypes* | Persistent symptoms n (%) | | | New-onset symptoms n (%) | | |
|---|---|---|---|---|---|---|
| | Cases n (%) | Controls n (%) | *P* values[1] | Cases n (%) | Controls n (%) | *P* values[1] |
| Myalgia | 14 (22.6%) | 7 (13.5%) | 0.2350 | 4 (6.5%) | 0 (0.0%) | 0,1242 |
| Hyposmia | 8 (12.9%) | 10 (19.2%) | 0.4420 | 3 (4.8%) | 2 (3.8%) | >0.9999 |
| Dysgeusia | 8 (12.9%) | 10 (19.2%) | 0.4420 | 1 (1.6%) | 0 (0.0%) | >0.9999 |
| Fever | 1 (1.6%) | 0 (0.0%) | >0.9999 | 0 (0.0%) | 0 (0.0%) | >0.9999 |
| Fatigue | 8 (12.9%) | 6 (11.5%) | >0.9999 | 12 (19.4%) | 6 (11.5%) | 0.3085 |
| Dry cough | 2 (3.2%) | 1 (1.9%) | >0.9999 | 5 (8.1%) | 1 (1.9%) | 0.2173 |
| Coryza | 4 (6.5%) | 0 (0.0%) | 0.1242 | 5 (8.1%) | 0 (0.0%) | 0.0618 |
| Dyspnea | 8 (12.9%)** | 0 (0%) | 0.0075 | 8 (12.9%)* | 1 (1.9%) | 0.0380 |
| Sore throat | 3 (4.8%) | 1 (1.9%) | 0.6242 | 3 (4.8%) | 1 (1.9%) | 0.6242 |
| Diarrhea | 2 (3.2%) | 0 (0.0%) | 0.4995 | 4 (6.5%) | 1 (1.9%) | 0.3737 |
| Headache | 14 (22.6%)* | 4 (7.7%) | 0.0390 | 10 (16.1%) | 3 (5.8%) | 0.1373 |
| Nausea/vomiting | 4 (6.5%) | 1 (1.9%) | 0.3737 | 9 (14.5%)** | 0 (0.0%) | 0.0037 |
| Loss of appetite | 0 (0.0%) | 0 (0.0%) | >0.9999 | 3 (4.8%) | 0 (0.0%) | 0.2492 |
| Abdominal pain | 0 (0.0%) | 1 (1.9%) | 0.4561 | 2 (3.2%) | 0 (0.0%) | 0.4995 |
| Expectoration | 0 (0.0%) | 0 (0.0%) | >0.9999 | 0 (0.0%) | 0 (0.0%) | >0.9999 |

Notes.

*Fisher's exact test, $p < 0.05$.

**$p < 0.01$.

## Part 2. Cognitive and psychiatric symptoms

Our study found that cases had a significantly higher rate of disability in daily activities, concentration complaints, anxiety, and "post-COVID sadness" compared to controls. In terms of numbers, the cases presented a rate of daily activities disability 4.7 times higher than that found in controls (45.2% *vs.* 9.6%; $P < 0.0001$) and difficulties in concentration 7.7 times higher (74.2% *vs.* 9.6%; $P < 0.0001$) (Table 7 and Fig. 6). Furthermore, the cases exhibited a twofold higher rate of complaints related to anxiety (66.1% *vs.* 34.6%; $P = 0.0013$) and "post-COVID sadness" (82.3% *vs.* 40.4%; $P < 0.0001$) compared to the control group. The ranking of frequency differed between cases and controls. Cases manifested mainly sadness, followed by concentration complaint, anxiety, and lastly, daily activities disability. Sadness was also the most prevalent phenotype in controls; however, it was followed by anxiety and then concentration complaint and daily activities disability. In other words, the control individual—who sleeps well and has no memory impairment—had relatively lower rates of disability in daily activities and concentration complaints (9.6% for both) compared to the cases that exhibited 45.2% and 74.2% of these phenotypes, respectively.

The results presented above suggest that cognitive and psychological phenotypes (*e.g.*, disability in daily activities, complaints of concentration difficulty, anxiety, and "post-COVID sadness") may emerge in conjunction with sleep and memory disturbances in cases experiencing long COVID, thereby forming phenotypic clusters of complaints. This observation carries substantial implications for comprehending the potential associations

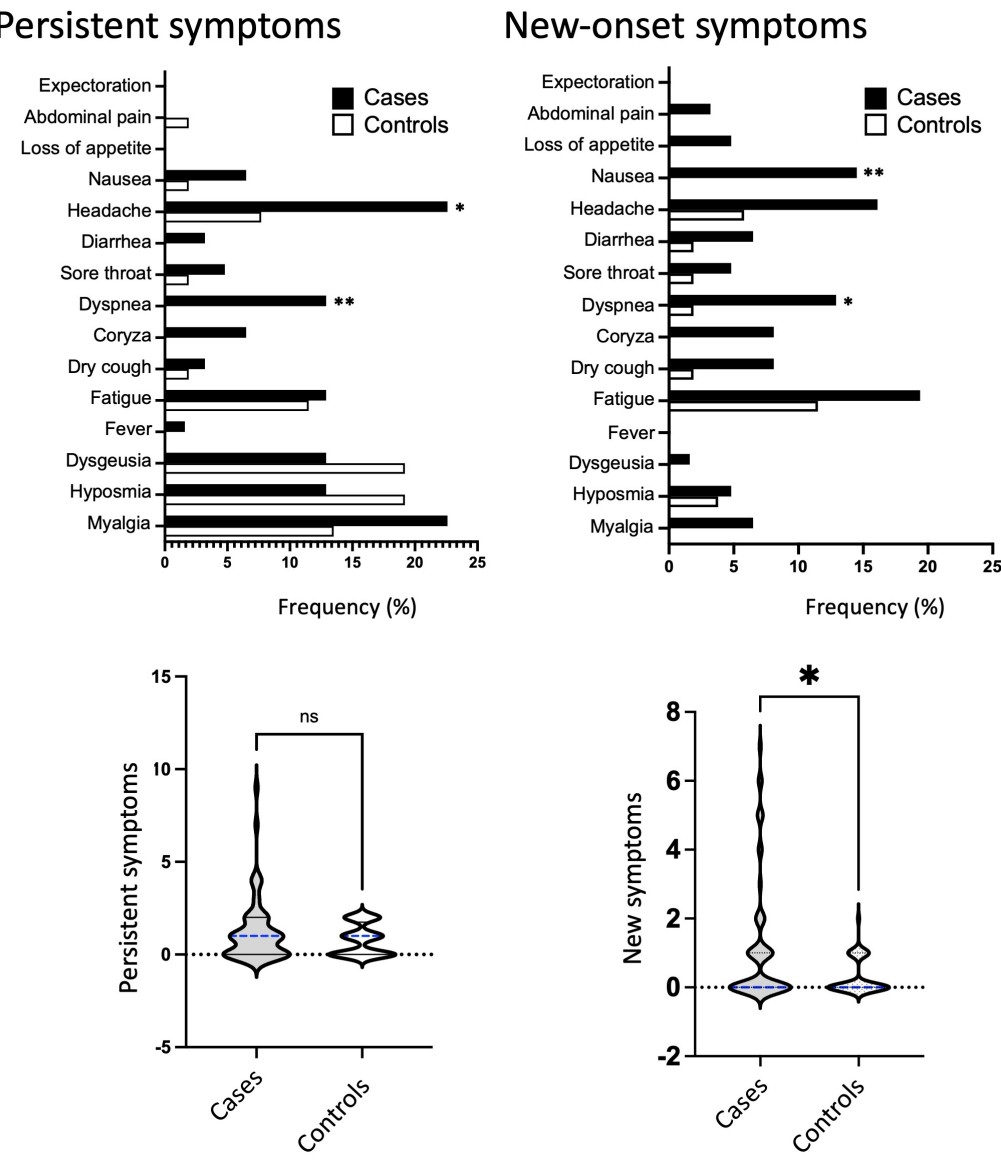

**Figure 4 Persistent and new-onset symptoms.** Frequency of persistent and new-onset symptoms in cases (represented by black bars) and controls (represented by white bars) during the long COVID phase. The bottom panel shows the number of persistent or new-onset symptoms and median values in cases (represented by gray violin) or controls (represented by white violin). Fisher's exact test was used to conduct statistical analysis, with significance indicated by $* p < 0.05$.

between sleep and memory disruptions and co-occurring conditions in long COVID patients. In our hypothesis, SARS-CoV-2 infection might have triggered a widespread brain pathology in individuals suffering from memory and sleep problems, *e.g.*, the cases. For testing this hypothesis, we organized individuals into the following experimental groups, according to the presence or absence of "post-COVID sadness" and anxiety: 1–Sadness without anxiety; 2–Sadness, with or without anxiety; 3–Anxiety without sadness; 4–Anxiety, with or without sadness; 5–Sadness with anxiety.

**Table 4  Categories of persistent symptoms.**

| Symptoms | Cases n (%) | Controls n (%) | P value[1] |
|---|---|---|---|
| With persistent symptoms (at least 1) | 37 (59.7%) | 28 (53.8%) | 0.5724 |
| Up to 2 persistent symptoms | 29 (46.8%) | 28 (53.8%) | 0.5729 |
| 3 or more persistent symptoms | 8 (12.9%)** | 0 (0.0%) | 0.0075 |

Notes.
** $p < 0.01$, Fisher's exact test.

**Table 5  Categories of new-onset symptoms.**

| Symptoms | Cases n (%) | Controls n (%) | P value[1] |
|---|---|---|---|
| With new symptoms (at least 1) | 28 (45.2%) | 14 (26.9%) | 0.0527 |
| Up to 2 new symptoms | 19 (30.6%) | 14 (26.9%) | 0.6843 |
| 3 or more new-onset symptoms | 9 (14.5%)** | 0 (0.0%) | 0.0037 |

Notes.
** $p < 0.01$, Fisher's exact test.

## Persistent symptoms

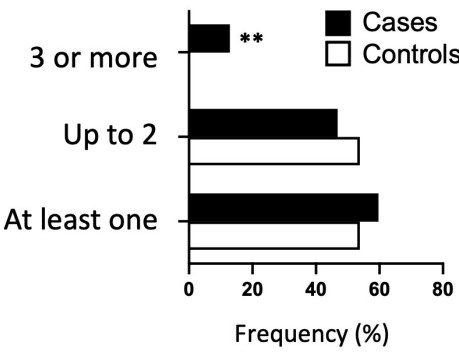

## New-onset symptoms

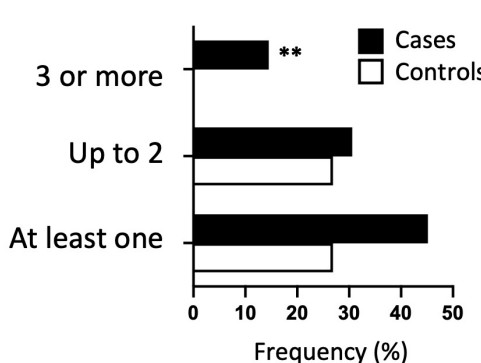

**Figure 5  Category of new-onset and persistent symptoms.** Frequency of persistent or new-onset symptoms categorized by the number of symptoms (0–1, 2, 3 or more) for cases (represented by black bars) and controls (represented by white bars). Fisher's exact test was used to conduct statistical analysis, with statistical significance indicated by ** $p < 0.01$.

As shown in Table 8, most cases ($n = 51$) showed sadness with or without anxiety (82.3%; group 2). Among them, most had sadness associated with anxiety ($n = 37$, 59.7%; group 5), and only a lower proportion presented sadness without anxiety ($n = 14$, 22.6%; group 1). Regarding controls, sadness, with or without anxiety, has affected a relatively lower number of individuals ($n = 21$, 40.4%; group 2) than cases. Half of these controls manifested sadness with anxiety ($n = 10$, 19.2%; group 5), and almost the same proportion had sadness without anxiety ($n = 11$, 21.2%, group 1).

**Table 6  Frequency of symptoms—long COVID.**

| Symptoms | Cases n (%) | Controls n (%) | *P* value |
|---|---|---|---|
| Total number of symptoms –Acute COVID[#1] | 238 (46.7%) | 272 (53.3%) | 0.6578 |
| Total number of symptoms –long COVID[#1] | 122 (64.3%)[*] | 68 (35.7%) | 0.0134 |
| Persistent symptoms[#1] | 64 (56.6%) | 49 (43.4%) | 0.3856 |
| New-onset symptoms[#1] | 58 (76.4%)[*] | 18 (23.6%) | 0.0108 |

Notes.

[*]$p < 0.05$, Mann–Whitney test.

[#1]Total frequency of symptoms was adjusted to account for the numerical difference of individuals between experimental groups of cases ($n = 62$) and controls ($n = 52$).

**Table 7  Cognitive and psychiatric phenotypes in long COVID.**

| Phenotype[1] | Cases n (%) | Controls n (%) | *P* value |
|---|---|---|---|
| Daily activities disability | 28 (45.2%)[****] | 5 (9.6%) | <0.0001 |
| Concentration complaint/sustained attention | 46 (74.2%)[****] | 5 (9.6%) | <0.0001 |
| Anxiety - Generalized Anxiety Disorder 2-item scale (GAD-2) score $\geq 3$ | 41 (66.1%)[**] | 18 (34.6%) | 0.0013 |
| Total "Post-COVID sadness" | 51 (82.3%)[****] | 21 (40.4%) | <0.0001 |

Notes.

[****]$p < 0.0001$.

[**]$p < 0.01$.

Fisher's exact test.

Our data thus revealed that sadness associated with anxiety, group 5, was more frequent in cases than controls (59.7% *vs.* 19.2%), and, even more, the rate of sadness associated with anxiety (group 5) regarding the sadness with or without the anxiety (group 2) was much higher in cases (37/51; 72.5%) than controls (10/21; 47.6%).

We finally evaluated if the occurrence of sadness was statistically associated with the occurrence of anxiety in cases and controls. As a significant study finding, we verified that sadness and anxiety were correlated in cases but not in controls ($P = 0.0212$; Spearman correlation test, blue and green bars in Fig. 7). The difference in the frequency of sadness associated with anxiety between cases and controls (59.7% *vs.* 19.2%) also showed statistical significance ($P < 0.0001$; Mann–Whitney's test, black bars in Fig. 7). The discovery that individuals who experience sleep and memory disturbances often display concurrent psychological phenotypes associated with long COVID supports the hypothesis that SARS-CoV-2 may contribute to a broad pathology that manifests as clusters of related brain phenotypes.

## DISCUSSION

In a previous research, our group found that 45% of individuals experienced sleep disturbances and "post-COVID sadness" 5–8 months after diagnosis (*Titze-de-Almeida et al., 2022*). The present case-control study reveals that individuals with sleep and memory disturbances, referred to as cases, exhibit a doubled rate of "post-COVID sadness"

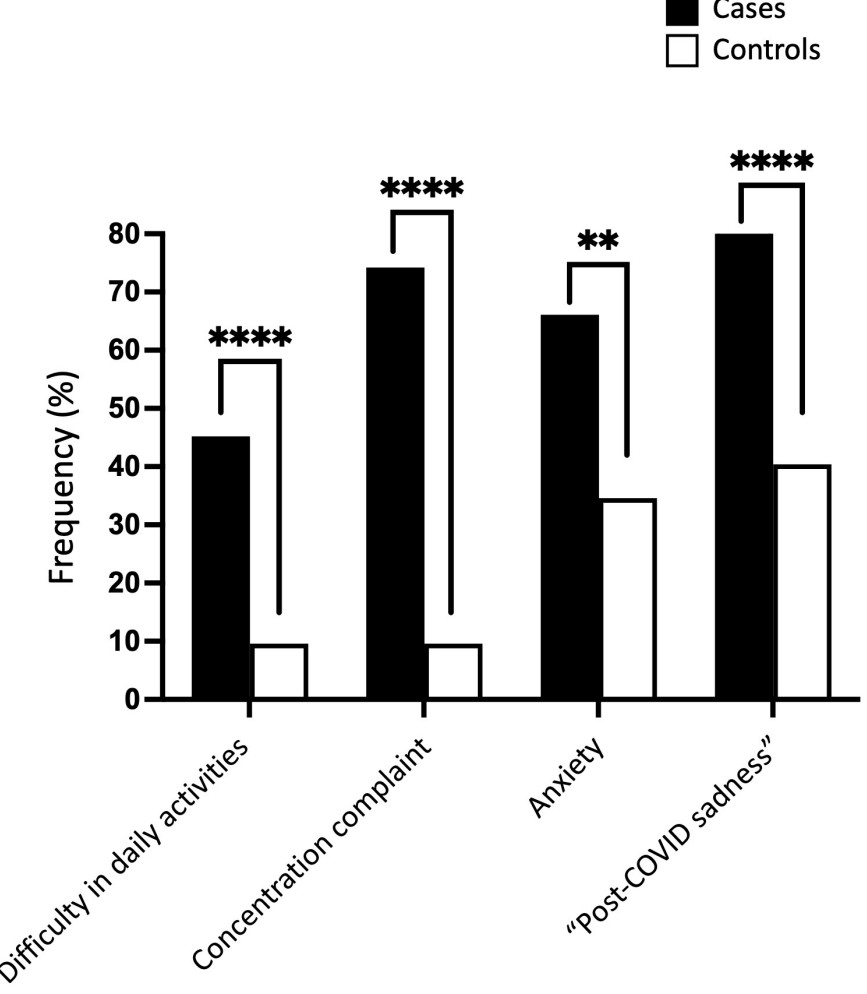

**Figure 6** **Cognitive and psychiatric phenotypes in cases and controls.** Frequencies of mental health complaints, including disability in daily activities, concentration disturbances, anxiety, and 'post-COVID' sadness, compared between cases (black bars) and controls (white bars) using Fisher's exact test. Statistical significance was indicated by *** $p < 0.001$ or ** $p < 0.01$.

**Table 8** **Frequency of sadness and anxiety in cases and controls.** The term sadness was used to represent "post-COVID sadness", for simplicity.

| Experimental Group | Cases n (%) | Controls n (%) |
|---|---|---|
| 1- Sadness without anxiety | 14 (22.6%) | 11 (21.2%) |
| 2- Sadness, with or without anxiety | 51 (82.3%) | 21 (40.4%) |
| 3- Anxiety without sadness | 4 (6.5%) | 8 (15.4%) |
| 4- Anxiety, with or without sadness | 41 (66.1%) | 18 (34.6%) |
| 5- Sadness with anxiety | 37 (59.7%) | 10 (19.2%) |

**Notes.**
The term sadness was used to represent "post-COVID sadness", for simplicity.

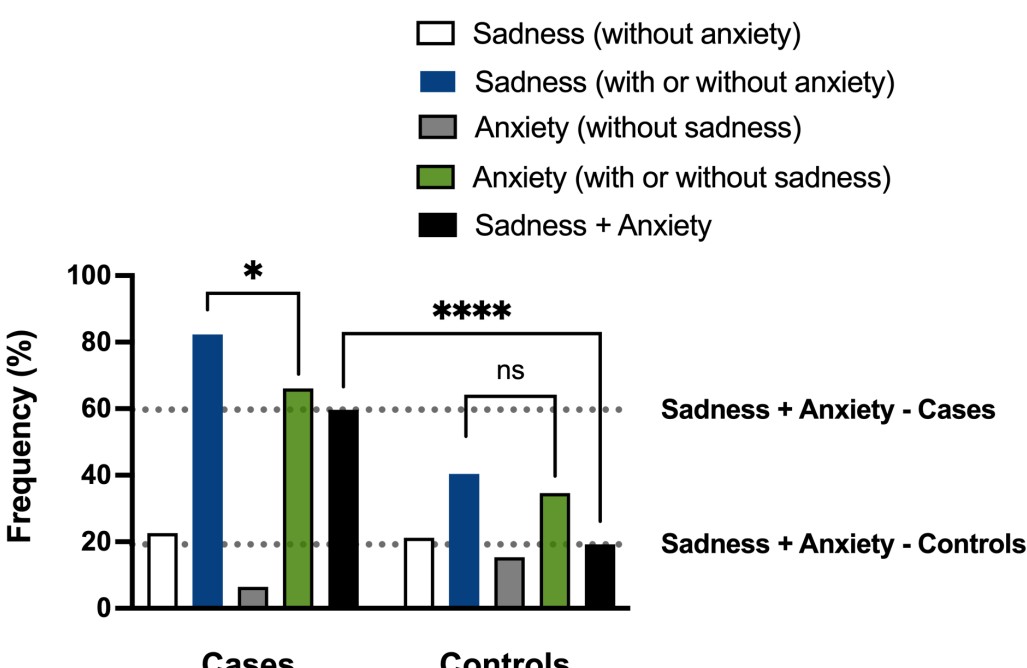

**Figure 7 Analyses of the sadness and anxiety phenotypes of cases and controls.** Frequencies of 'post-COVID sadness' and anxiety in cases (individuals with sleep and memory disturbances) and controls (without both complaints). Participants were grouped based on their symptoms, and data were analyzed using Spearman correlation and Mann–Whitney tests. A significant correlation was found between sadness and anxiety in cases (*$P = 0.0212$, Spearman correlation test) but not in controls (ns, non-significant). Furthermore, a significant difference was observed in the frequencies of co-occurring sadness and anxiety between cases and controls (****$P < 0.0001$, Mann–Whitney test).

compared to controls (82.3% *vs.* 40.4%). Our COVID-19 data corroborated the well-known relationships between sleep disorders and depressive symptoms (*Efstathiou et al., 2022*; *Ustun, 2021*). In general, the association between insomnia and depression is so common that some authors suggest that, in the absence of insomnia symptoms, the diagnosis of depression should be approached with caution (*Luca, Luca & Calandra, 2013*). Additionally, chronic insomniacs with depression/anxiety have been reported to exhibit reduced emotional reactivity (emotional blunting) in tests of facial recognition for expressions of anger or sadness (*Kyle et al., 2014*).

Sleep disorders in the context of long COVID-19 can not only be frequent but can also have a negative long-term impact on the outcome of COVID-19 (*Schilling, Meyer-Lindenberg & Schweiger, 2022*). A systematic review aimed at determining the prevalence of neurological and neuropsychiatric symptoms reported 12 or more weeks after the acute onset of COVID-19 in adults analyzed 1458 articles (a total of 11,324 patients), revealing memory problems in 27%, attention disorders in 22%, sleep disorders in 31%, anxiety in 23%, and depression in 12% of patients (*Premraj et al., 2022*). The same study linked hospitalization to a higher frequency of memory problems (OR 1.9, 95% CI [1.4–2.3]). Furthermore, the prevalence of manifestations such as sleep disorders, anxiety, and depression increased significantly over time. Our previous work also showed that memory

disturbances were correlated with sleep complaints and sadness during long COVID (*Titze-de-Almeida et al., 2022*). The rates of sleep complaints and sadness were respectively 27.1% and 27.5% in all study individuals but reached more than double, 68.1% and 69.1%, in the subgroup reporting memory problems. This data suggests that all these phenotypes might have occurred in the same individuals who might have the same affected brain areas. In the present study, we found that anxiety and sadness correlated significantly only if individuals were cases, *i.e.,* individuals with sleep and memory problems, corroborating other studies that found relevant proportions of sleep disturbances and psychiatric symptoms (*Ahmed et al., 2021*; *Barros et al., 2020*; *Deng et al., 2021*; *Morin et al., 2021*; *Passos et al., 2020*).

Previous studies have shown that several other psychiatric disorders and quality of life problems are common in individuals who have survived COVID-19, which reinforces the hypothesis of the involvement of different brain areas. A self-assessment study with 402 individuals showed a prevalence of 28% for post-traumatic stress disorder, 31% for depression, 42% for anxiety, 20% for obsessive-compulsive disorder symptoms, and 40% for insomnia (*Mazza et al., 2020*). Another study, a longitudinal cohort investigation, followed 1,119 individuals who survived hospitalization with COVID-19 and were discharged in Wuhan-China between January 7th and May 29th, 2020, concluded that survivors with long COVID-19 symptoms had lower health-related quality of life, worse exercise capacity, more mental health abnormalities, and higher healthcare utilization after discharge than survivors without long COVID-19 symptoms. In addition, they had more prevalent symptoms of anxiety and depression compared to the control group during the same 2-year follow-up period of the study (*Huang et al., 2022*).

Deficits in cognitive function are a relevant issue after SARS-CoV-2 infection. A study with a large group of individuals ($n = 81,337$), predominantly British, who recovered from COVID-19 with or without reports of symptoms, showed significant cognitive deficits in patients who were or were not hospitalized in the acute phase of the disease (*Hampshire et al., 2021*). Regarding cognitive impairment, a meta-analysis indicated that 22% of individuals exhibited cognitive impairment 12 or more weeks after the diagnosis of COVID-19 (*Ceban et al., 2022*). Our study indirectly addressed cognition by evaluating impairments in routine daily activities and level of concentration during long COVID. Compared to controls, cases showed increased complaints in daily activities (45.2% *vs.* 9.6%) and concentration/sustained attention (74.2% *vs.* 9.6%).

The results of the present study showed that patients with memory disorders, as well as those with dissatisfaction with their sleep associated with long COVID, had a significant increase in anxiety and "post-COVID sadness" scores when compared to controls, 66.1% *vs.* 34.6%, and 82.3% *vs.* 40.4%, respectively. There are several hypotheses to explain the correlation between sleep disorders and mood changes in post-COVID syndrome. Among them, the inflammatory phenomenon and the negative psychosocial impact of the pandemic on patients' lives should be considered. Concerning neuroinflammatory process, cytokines TNF-α (tumor necrosis factor) and IL-6, which are upregulated in COVID-19, are also present in major depressive disorder and Alzheimer's disease. This suggests that such cytokines may generate dysfunctional stress-related responses, mood alterations, and cognitive impairment in COVID-19 survivors (*Lyra et al., 2022*). Despite the mechanisms

of this interaction still being studied, they are probably multifactorial (*Efstathiou et al., 2022*).

Regarding mood disorders, functional brain imaging studies in groups of patients with major depressive disorder showed states of hypofunction in areas such as the prefrontal cortex and anterior cingulate gyrus, associated with active thoughts of sadness or pessimism (*Wostyn, 2021*). Significant metabolic alterations and even volume reductions were also described in other equally important regions, such as the temporal lobes and basal ganglia, in this context (*Brody et al., 2001*). It is interesting to note that many of these studies were conducted using sleep deprivation as an ''activator'' of mood changes (*Wostyn, 2021*). Similarly, a systematic review study suggests that inflammatory mechanisms of SARS-CoV-2 infection may also generate or exacerbate cognitive impairment (*Rosenblat et al., 2014*), with the causal relationship between specific proinflammatory cytokines, mood alterations, and cognitive decline being a relatively well-established association (*Ceban et al., 2022*).

The present research has some important limitations. The cases and controls study design used in this research may introduce memory bias due to administering the questionnaires at different times. Additionally, the small sample size is justified by various factors. Originally, the sample consisted of 362 individuals who were contacted *via* telephone during the acute phase of COVID-19, at the peak of the pandemic. In our country, some individuals lack motivation to participate in scientific studies, while others agreed to take part only after establishing an emotional connection with the researcher. Unfortunately, a portion of the individuals ($n = 126$) declined to participate in the long COVID study. Another important aspect of our study is that the majority of participants ($n = 122$) in the initial cohort of 362 individuals had a mild disease presentation, did not require hospitalization, and fully recovered after the acute phase. Consequently, they did not experience persistent or new-onset symptoms, which means they did not meet the WHO criteria for a diagnosis of long COVID. It is also noteworthy that the sample size only represents two hospitals in the state, and therefore may not be representative of the entire population of Distrito Federal, Brazil.

Our study found that cases presented more symptoms than controls in long COVID, including higher rates of new-onset and persistent symptoms such as headache and dyspnea. These results indicate that SARS-CoV-2 infection may trigger widespread brain pathology that ultimately dysregulates the physiology of different systems. Hence, a potential bias could arise if the number of symptoms has affected the results rather than the sleep and memory phenotypes that formed the study groups. However, we consider that this potential limitation was not significant, taking into account the following factors. Firstly, the vast majority of experimental individuals were non-hospitalized, which implies that they developed a milder form of the disease without intubation or ICU requirements. Only seven cases (11.3%) and six controls (11.5%) needed oxygen therapy and ICU. During the acute phase of the disease, cases and controls presented the same prevalent clinical signs: myalgia, hyposmia, dysgeusia, and headache, with no statistical difference between the groups for these symptoms or others. In the long COVID phase, differences were observed between cases and controls, but only for 4 out of the 15 evaluated symptoms (approximately

27% of the symptoms showed variation, meaning that there was no significant difference in 73% of the evaluated symptoms). The difference in long COVID was observed for coryza, dyspnea, headache, and nausea/vomiting. However, some of these phenotypes affected a relatively small percentage of cases and controls. For example, coryza affected nine cases (14.5%) *versus* 0 controls, dyspnea affected 16 cases (25.8%) *versus* one control (1.9%), and nausea/vomiting affected 13 cases (21%) *versus* one control (1.9%). The difference was larger for headache, with 24 cases (38.7%) *versus* seven controls (13.5%). We would like to point out that for two phenotypes, controls were slightly more symptomatic than cases: hyposmia and dysgeusia. Regarding persistent symptoms, cases and controls differed significantly only for two out of 15 symptoms: dyspnea and headache, which affected eight and 14 cases *versus* 0 and four controls, respectively. For new-onset symptoms, only two differed significantly: dyspnea and nausea/vomiting, affecting eight and nine cases *versus* one and zero controls, respectively. In summary, both groups do not show significant differences in COVID-19 symptoms during the acute phase, and cases only presented an increase in a relatively small number of phenotypes, which, in turn, affected a relatively small number of individuals.

However, the matter of disease severity and its consequences is truly complex and influenced by multiple variables, as demonstrated in the study conducted by *Zeng et al. (2023)*. The severity of the disease may influence the outcomes, but other variables such as age, gender, and living in high-income countries also play a significant role (*Zeng et al., 2023*). In fact, individuals with a milder disease presentation can experience mental burdens, including anxiety and memory impairment. Supporting this notion, *Magnusdottir et al. (2022)* found that individuals who were bedridden for 7 days or more during the acute illness phase (22% of the total) showed persistently high levels of depression and anxiety symptoms compared to individuals who were never bedridden due to COVID-19. The latter group presented lower risks of mental morbidities (*Magnusdottir et al., 2022*).

In contrast, other studies have found no correlation between disease severity and the phenotypes examined in this study. For instance, *Stavem et al. (2022)* reported a decline in short-term memory, visuospatial processing, learning, and attention at 11 months post-COVID-19 infection in a cohort of non-hospitalized patients ($n = 234$) (*Stavem et al., 2022*). However, the study was unable to stablish a correlation between symptom severity and the extent of mental impairment. Similarly, *Badenoch et al. (2022)* analyzed 51 studies involving 18,917 patients and found that sleep disturbance affected 27.4% of individuals. This study also did not discover substantial evidence linking persistent symptoms (such as sleep disturbances, fatigue, objective cognitive impairment, anxiety, and post-traumatic stress) to the initial severity of COVID-19 infection. Our Brazilian cohort exhibited similar findings, as no significant differences were observed between the cases and controls during the acute phase of COVID-19. The distinctions became evident only 5 to 8 months after the diagnosis, specifically during the period of long COVID.

COVID-19 in Brazil has emerged in successive waves of SARS-CoV-2 genetic variants that have spread across this vast South American country (*Alcantara et al., 2022*). In our study, COVID-19 infections were observed in both cases and controls between September and December 2020, when different lineages were co-circulating. For instance, in late 2020,
variants B.1.1.28 and P.2 (zeta) were prevalent in various regions of Brazil, including the Distrito Federal where our study took place. P.2 evolved from the B.1.1.28 variant, which emerged in December 2020, and became prevalent in 2021 along with the Gamma/P.1 variant (*Giovanetti et al., 2022*). In our cohort, infections were observed in both cases and controls in proportions that did not show any statistical differences between the months of September to December 2020. However, the study did not conduct an examination of the SARS-CoV-2 variants. Hence, it is not possible to infer whether there were any differences in the frequency of the B.1.1.28 or P.2 (zeta) variants between the cases and controls. This limitation should be considered when interpreting the findings of the present study.

The etiology of long COVID is likely multifactorial, involving various pathogenic mechanisms (*Crook et al., 2021*; *Davis et al., 2023*). SARS-CoV-2 particles are usually eliminated from the body within weeks of infection, but persistent fragments of the RNA genome and spike proteins may play a role in triggering pathological processes associated with long COVID (*Griffin, 2022*; *Marshall, 2021b*; *Swank et al., 2022*). In this research, cases had a significantly lower vaccination ratio compared to controls (30.7% *vs.* 51.9%). Although the current study was not designed to examine the effect of vaccination after COVID-19 infection, our data suggest that a lower vaccination rate in cases might have contributed to a more symptomatic form of long COVID. Similarly, vaccination may potentially benefit controls in developing a milder form of long COVID. The impact of vaccination on preventing long COVID outcomes is highly significant and promising, warranting further investigation (*Byambasuren et al., 2023*; *Tran et al., 2023*; *Watanabe et al., 2023*).

It is important to note that we did not use a specific tool to diagnose depression in individuals who reported "sadness post-COVID". Thus, comparison of our results with existing literature on this specific aspect may be limited. However, we must also consider the challenges faced during data collection due to the COVID-19 pandemic. Gathering a large number of symptoms, including the classic COVID-19 symptoms, was extremely difficult, given the high levels of distress among the population and the limited level of education, as we collected data from a lower income cohort of subjects in Brazil. To ensure the accuracy of the data collected, we implemented a comprehensive training process. All researchers completed the online course 'COVID-19 contact tracing' (https://www.coursera.org/learn/covid-19-contact-tracing). Subsequently, they received careful training and supervision from senior researchers, tutors, and physicians to conduct phone interviews and collect data from the enrolled participants.

In this final part of discussion, we will address the consequences of SARS-CoV-2 infection regarding co-occurrence of phenotypes in long COVID. In our first study, we observed that a significant proportion of the individuals experienced memory complaints, anxiety, "post-COVID sadness", and sleep problems, with rates of 40%, 37%, 45%, and 46%, respectively (*Titze-de-Almeida et al., 2022*). Notably, those who reported memory complaints also had higher rates of sleep problems and "post-COVID sadness", suggesting a potential biological link between different brain areas affected by the virus. This link may have contributed to interconnected cognitive and emotional symptoms.

The relation between sleep and memory problems was also reported previously, in a study that also include non-hospitalized individuals (76%) with a mean age of 46 years that was similar to our cohort. This study found that perceived sleep concerns predicted self-reported memory disturbances, whereas the severity of posttraumatic stress disorder symptoms or anxiety symptoms did not significantly predict cognitive impairment or self-reported memory disturbances (*Brown et al., 2022*).

In a systematic review and meta-analysis of 51 studies involving 18,917 patients, sleep disturbance was the most prevalent neuropsychiatric symptom, affecting 27.4% of the individuals. Fatigue (24.4%), objective cognitive impairment (20.2%), anxiety (19.1%), and post-traumatic stress (15.7%) were also frequently reported (*Badenoch et al., 2022*). The authors did not find significant evidence linking these persistent symptoms to the severity or duration of the initial COVID-19 infection. Additionally, a multicenter study conducted in Spain with 1,142 individuals at 7 months post-COVID-19 diagnosis found that 34.5% experienced poor sleep quality, followed by depressive symptoms (19.7%) and anxiety (16.2%) (*Fernandez-de-Las-Penas et al., 2021*). A nationwide study conducted in Denmark with 61,000 COVID-19-positive individuals and 92,000 negative individuals found that 10.9% experienced sleep problems. Sleep problems ranked among the top three most common issues in COVID-19-positive individuals, along with fatigue/exhaustion (11.1%) and dysgeusia (9.8%) (*Sorensen et al., 2022*). Moreover, sleep problems showed the highest adjusted risk differences (RD), RD = 10.92%, followed by dysgeusia (RD = 8.68%), and fatigue/exhaustion (RD = 8.43%) among the 21 symptoms examined.

The current case-control study not only validates our previous findings on long COVID symptoms but also provides confirmation of the presence of co-occurring psychological phenotypes in individuals experiencing sleep and memory problems. These phenotypes include anxiety and 'post-COVID sadness.' This finding strengthens the existing evidence that SARS-CoV-2 infection leads to interconnected phenotypes that manifest as clusters of symptoms.

Two other studies have also reported clusters of symptoms in COVID-19 patients. *Evans et al. (2021)* identified four clusters of recovering phenotypes at six months after hospital discharge, focusing on mental health, physical performance, and cognition impairments (*Evans et al., 2021*). Their study included hospitalized patients and did not specifically examine memory and sleep problems. However, it underscored the importance of stratifying patients and addressing a broader range of health conditions in medical interventions. Our study drew inspiration from the work of *Kenny et al. (2022)*, which also identified symptom clusters (*Kenny et al., 2022*). The majority of individuals in their study were women, and their initial illness was mild, similar to our Brazilian cohort. Kenny et al. focused on specific symptoms that grouped into three main clusters. Cluster 1 primarily comprised "pain symptoms" like joint pain, myalgia, and headache. Cluster 2 included cardiovascular symptoms such as chest pain, shortness of breath, and palpitations. Finally, Cluster 3 presented fewer symptoms overall. They used multiple correspondence analysis (MCA) for clustering and employed heat-mapping as an innovative method to validate the symptomatic distribution. The authors concluded that such symptom clusters may represent distinct pathogenic mechanisms underlying COVID-19 sequelae.

In summary, to effectively address mental disturbances associated with long COVID, it is crucial to raise awareness among public health services regarding the clinical manifestations of clustered symptoms. These symptoms include depressive mood and anxiety in individuals experiencing sleep and memory disturbances. Such symptoms may indicate a widespread brain pathology associated with SARS-CoV-2. Recognizing these manifestations calls for comprehensive clinical examinations and appropriate clinical management. Additionally, further studies are necessary to better understand the causes of sleep and memory disorders in long-term COVID, especially considering the mutability of the virus and the potential neuroprotective effect of vaccination.

## CONCLUSIONS

Our study highlights that individuals experiencing sleep and memory problems do not demonstrate significant differences in symptomatology during acute COVID-19. However, they do experience an increased occurrence of headache and dyspnea during long COVID. Furthermore, the concurrent emergence of sadness and anxiety among those with sleep and memory concerns reinforces the idea that SARS-CoV-2 infection induces extensive brain pathology, which is manifested through clusters of interconnected phenotypes. The complexity of the mental burden caused by SARS-CoV-2 and the promising neuroprotective effect of vaccination against long COVID are issues that require significant efforts from the scientific community and special attention from governments worldwide.

## ACKNOWLEDGEMENTS

The authors extend their sincere appreciation to all the institutions and individuals who participated in this study, including those who were diagnosed with COVID-19. Special recognition is given to the scientists from the 'Research Center for Major Themes – COVID-19 group', who provided vital assistance in data collection, scientific discussions, and administrative tasks throughout the study: Adriana Pinheiro Ribeiro, Sabrina Simplício de Araujo Romero Ferrari, Thaylise Ramalho da Cunha, Letícia Dias dos Santos Silva, Clarisse Santos Ferreira, Caroline Pena Silva, Julia Teixeira Silva, José Eduardo Lemes da Silva, Sthéfani Alvares da Costa Moura, Ingrid Fernandes da Rocha, Jade Kemp Wanderley, Milena Henrique Gomes, Gyulyanna Siqueira Lima, Isadora Magalhäes Cunha, Brenda Lopes da Silva, Kétlen Monique Hoch Barbosa, Larissa Cristina de Souza Akiyama, Rafael Torres Serpa, Nayane Karoline França da Fonseca, Thales Leone Corrêa, Paula Vincunā Silva Neves, Victória Gabriella Campos de Jesus, Marcela Lopes Alves, Mariana Moura Moutinho, Amanda Kelly Costa de Carvalho, Amanda Sodré Almeida, Ana Luiza Alves Ferreira, Andréia da Costa Azevedo, Menezes, Beatriz Pereira da Rocha Lima, Caroline Silva de Oliveira, Deborah Luisa Amorim Silva, Diego Fonseca Oliveira Bispo, Emanoelle Castro Ribeiro, Eric Ferreira Santana, Flaviane de Souza Brito, Gabriela da Silva Freire, Isabela Alfredo Vaz, Jamila Santos Khalifa, Jéssica Moreira da Silva, Layanne Gomes Araujo, Let.cia de Oliveira Mayer, Lorena de Sousa Aires, Luciana Antunes de Faria, Maria Fernanda Pereira Neves Leite Silva, Maria Isabel Burbano Sandoval, Mariana Souza Santos, Rebecca Conceição de Freitas, Sofia Rocha Santos Quaresma, Talita Fernandes Nunes,

Thaís Lorrane de Melo Silva, Yago Lucas da Silva, Cynthia Yara da Silva Ribeiro, Suamir Jorge de Azevedo Campos, Lucas Luiz Vieira, Fabiano José Queiroz Costa.

### Funding
This work was supported by the Ministry of Education (MEC) (TED/MEC n. 9249) and The National Council for Scientific and Technological Development (CNPq). The funders had no role in study design, data collection and analysis, decision to publish, or preparation of the manuscript.

### Grant Disclosures
The following grant information was disclosed by the authors:
The Ministry of Education (MEC): TED/MEC n. 9249.
The National Council for Scientific and Technological Development (CNPq).

### Competing Interests
The authors declare there are no competing interests.

### Author Contributions
- Ricardo Titze-de-Almeida conceived and designed the experiments, analyzed the data, prepared figures and/or tables, and approved the final draft.
- Pedro Henrique Araújo Lacerda conceived and designed the experiments, performed the experiments, prepared figures and/or tables, and approved the final draft.
- Edson Pereira de Oliveira conceived and designed the experiments, performed the experiments, prepared figures and/or tables, and approved the final draft.
- Mariah Eduarda Ferreira de Oliveira conceived and designed the experiments, performed the experiments, prepared figures and/or tables, and approved the final draft.
- Yngrid Sallaberry Silva Vianna conceived and designed the experiments, performed the experiments, prepared figures and/or tables, and approved the final draft.
- Amanda Machado Costa conceived and designed the experiments, performed the experiments, prepared figures and/or tables, and approved the final draft.
- Eloísa Pereira dos Santos conceived and designed the experiments, performed the experiments, prepared figures and/or tables, and approved the final draft.
- Louise Marie Coelho Guérard conceived and designed the experiments, performed the experiments, prepared figures and/or tables, and approved the final draft.
- Matheus Augusto de Miranda Ferreira conceived and designed the experiments, performed the experiments, prepared figures and/or tables, and approved the final draft.
- Isabel Cristina Rodrigues dos Santos conceived and designed the experiments, performed the experiments, prepared figures and/or tables, and approved the final draft.
- Jéssica Daniele da Silva Gonçalves conceived and designed the experiments, performed the experiments, prepared figures and/or tables, and approved the final draft.

- Gabriel Ginani Ferreira conceived and designed the experiments, authored or reviewed drafts of the article, and approved the final draft.
- Simoneide Souza Titze-de-Almeida conceived and designed the experiments, analyzed the data, authored or reviewed drafts of the article, and approved the final draft.
- Pedro Renato de Paula Brandão conceived and designed the experiments, analyzed the data, authored or reviewed drafts of the article, and approved the final draft.
- Helena Eri Shimizu conceived and designed the experiments, analyzed the data, authored or reviewed drafts of the article, and approved the final draft.
- Andrezza Paula Brito Silva conceived and designed the experiments, analyzed the data, authored or reviewed drafts of the article, and approved the final draft.
- Raimundo Nonato Delgado-Rodrigues conceived and designed the experiments, analyzed the data, authored or reviewed drafts of the article, and approved the final draft.

## Human Ethics

The following information was supplied relating to ethical approvals (i.e., approving body and any reference numbers):

In accordance with ethical standards, approval was obtained from the ethics committee of the Institute of Strategic Health Management of the Federal District (IGESDF) with CAEE number 36147920.1.0000.8153. Informed consent was obtained from all individual participants included in the study.

## Data Availability

The raw data are available in the Supplemental File.

## Supplemental Information

Supplemental information for this article can be found online at http://dx.doi.org/10.7717/peerj.16669#supplemental-information.

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
