# Peer review of "Sleep and memory complaints in long COVID: an insight into clustered psychological phenotypes"

_PeerJ, doi:10.7717/peerj.16669_

## Round 0.1 · original submission · Major Revisions

Your manuscript has now been seen by two reviewers. You will see from their comments below that while they find your work of interest, some important points are raised. In particular, the choice of focusing on sleep and memory requires stronger motivation, justification and discussion. We are interested in the possibility of publishing your study, but would like to consider your response to these concerns in the form of a revised manuscript before we make a final decision on publication. We therefore invite you to revise and resubmit your manuscript, taking into account the points raised. Please use clear, unambiguous, technically and grammatically correct English throughout. Please highlight all changes in the manuscript text file.

Reviewer 1 ·

Basic reporting

There are multiple examples of incorrect usage throughout the text. For example:
- "relevant proportion" on line 49
- "referred two times more" (lines 393-394).

I think the language should be edited for simplicity and conciseness, as well as technical correctness.

There are a few incorrect citations: eg. on line 87: "Writing Committee for the et al". This seems to be an incorrect citation to a corporate author.

Experimental design

At no point do the authors make it clear why they chose these groups. Why are sleep and memory difficulties classed together while selecting the case group? Why are these two symptoms (and not any others, eg fatigue or anxiety) taken together?

There is hardly any description of the methods of recruitment or selection. Were only adults recruited? What was the average age of the two groups? What proportion required intensive care during acute COVID? What proportion had been vaccinated at the time of ascertainment? Were symptoms first identified in 2020, 2021 or 2022, when different variants were in circulation, and the symptom presentations showed substantial variation? Were any of the subjects in treatment for their long COVID symptoms?

Validity of the findings

This is the most significant area of concern.

I have concerns both about the internal validity of this study (relating to the classification of cases and controls) as well as the generalizability of these findings.

Firstly, there appears to be a systematic classification of patients with higher symptom counts into the case group. Intuitively, this seems likely as the classifier itself is the presence of two specific symptoms of long COVID, and the use of these two symptoms over any other two, is not justified.

Given that the cases and controls are selected on this very basis (of having two additional symptoms, namely sleep and memory difficulties), this seems a logical conclusion. I would encourage the authors to look up the psychiatric literature on medically unexplained physical symptoms, which clearly show that in primary care patients (whom these patients mostly resemble in their presentation, with whatever little information is presented) symptoms show a bimodal distribution, with a number of patients who have higher levels of morbidity, and a number of mutually exlusive physical symptoms. It is likely that the two groups selected here are centred on the two peaks of this bimodal distribution, and all results arise from this finding.

Secondly, it seems commonsensical to assume that subjects with a larger number of complaints would also experience a greater frequency of anxiety and depressive symptoms, as well as poorer functioning. It is unclear what nuance or additional information is provided by the results of this study, beyond this notion. Moreover, there is a robust literature on the relationship between physical symptoms (of any etiology) and their relationship with common mental disorders (including anxiety, depression and insomnia) on the one hand, and with cognitive dysfunction (including memory deficits) on the other hand.

Thirdly, the authors have based their studies exclusively on telephonically ascertained, subjective complaints. Except for the GAD-2 (an ultrabrief screener for community use), it is unclear whether any of the questionnaires that were used were validated for this purpose, and what, if any, precautions were taken to ensure that the reporting was valid. In the absence of these, this is a major limitation that the authors should consider in their discussion, and should make all the findings quite tentative.

The analytical method and the discussion are quite haphazard, and would benefit greatly from extensive editing. I would advise rather than adopt multiple methods of analysis, the authors should focus on one central, major conclusion that they would like to make from this study, and draw attention to that finding both in the review and in the discussion. After reading the article, I am unable to make out whether sleep/memory symptoms are being suggested as "biomarkers", or as screening questions. Both points are made at different points in the report, and the analytical method chosen (including the ACGT framework) are not appropriate to either of these.

If the aim was to identify clustering patterns of dysfunction associated with sleep/memory deficits, this effort does not find any justification in the review, nor does it find much mention in the discussion.

Additional comments

The reporting of this article does not adhere to ordinary conventions. The methods and results section are not in enough detail to allow replicability, nor to comment on the generalizability of results.

There is a lack of coherence between the various analytical methods chosen (correspondence, heat mapping etc) and these are not discussed with enough detail for the readers to identify key findings.

Reviewer 2 ·

Basic reporting

This manuscript by Almeida et al. aimed to assess the clinical features of individuals diagnosed with Long COVID (5-8 months post-diagnosis) who reported sleep and memory problems (62 cases), comparing them to a control group without these issues (52 controls). The results indicated that cases were more likely to experience coryza, dyspnea, headache, and nausea/vomiting. In terms of new-onset symptoms, 12.9% of cases reported dyspnea, and 14.5% experienced nausea/vomiting, while in the control group, these numbers were only 1.9% and 0.0%, respectively. Furthermore, cases exhibited significantly higher rates of persistent headaches (22.6% vs. 7.7%) and dyspnea (12.9% vs. 0.0%), along with a range of mental health complaints. These included disability in daily activities (45.2% vs. 9.6%; referred to as phenotype A for simplicity), concentration/sustained attention difficulties (74.2% vs. 9.6%; referred to as phenotype C), anxiety - GAD-2≥3 (66.1% vs. 34.6%; referred to as phenotype G), and total post-COVID sadness (82.3% vs. 40.4%; referred to as phenotype T). By establishing a binary ACGT signature, based on the presence or absence of these phenotypes (1 or 0, respectively), a distinct 'G1T1' cluster emerged, characterized by individuals experiencing anxiety and post-COVID sadness. Notably, cases were three times more prevalent than controls within this cluster (59.7% vs. 19.2%). Moreover, the statistical analysis revealed that sadness was significantly associated with anxiety only among the cases. Multiple correspondence analyses demonstrated that cases and controls formed distinct clusters based on their ACGT phenotypes. These findings strongly suggest that sleep and memory disturbances may serve as valuable biomarkers for identifying Long COVID individuals with widespread brain pathology associated with SARS-CoV-2, warranting comprehensive neuropsychiatric evaluations. Overall, the manuscript by Almeida et al. provides valuable insights into the clinical features of Long COVID. However, there are several areas that require minor revisions to enhance the quality and comprehensibility of the article.

Minor revisions:
1) First, the authors should address the small sample size issue, as the study included only 62 cases and 52 controls. Increasing the sample size would strengthen the generalizability of the findings.

2) Second, it is important for the authors to clarify the specific duration of Long COVID (5-8 months) in relation to the diagnosis, providing a clearer understanding of the study population.

3) Third, sentence structure could be improved for better clarity and readability. Some sentences could be simplified or rephrased for clarity and readability. For instance, instead of "Regarding new-onset symptoms, 12.9% of cases had dyspnea, and 14.5% experienced nausea/vomiting, whereas in the control group, there were only 1.9% and 0.0%, respectively,". The authors could rephrase it as "New-onset symptoms, such as dyspnea and nausea/vomiting, were reported by 12.9% and 14.5% of cases, respectively, compared to only 1.9% and 0.0% in the control group."

4) Fourth, additional participant information, such as demographics and relevant comorbidities, should be included to provide a more comprehensive understanding of the study population.

5) Fifth, the authors should explicitly highlight the limitations of the study, such as the small sample size and potential selection bias, to address any factors that may impact the interpretation of the findings.

6) Sixth, the discussion section should focus on the potential implications of the study findings for clinical practice, public health, and further research in the field of Long COVID and its associated symptoms.

7) Seventh, thorough proofreading is necessary to address any grammar, spelling, and punctuation errors that may affect the clarity and accuracy of the article.
7.1) Page 6, line 60: Remove the word "viral"; the acronym SARS-CoV-2 already encompasses the term virus, as it stands for Severe Acute Respiratory Syndrome Coronavirus 2.

7.2) Page 8, line 156: Provide a description of the acronym GAD-2, which stands for Generalized Anxiety Disorder 2-item scale (GAD-2).

7.3) Page 9, line 171 and page 13, line 350: How did the authors precisely evaluate post-COVID sadness and anxiety? It is essential to consider that individuals can experience sadness or anxiety for numerous reasons, including financial difficulties, relationship issues, work conflicts, unemployment, or even waking up in a low mood. How can we effectively distinguish these factors from the specific effects attributed to Long COVID? This aspect appears to be a significant limitation within the article.

7.4) Page 16, lines 458 to 476: These sentences are purely speculative. The current study did not assess cytokines; therefore, it is not possible to establish this correlation.

7.5) Additionally, the authors should explicitly state the limitations of the study, such as the small sample size, potential selection bias, or any other factors that may impact the interpretation of the findings.

8) The authors should point out the differences of this study in relation to the following published works:
8.1) Titze-de-Almeida R, da Cunha TR, Dos Santos Silva LD, Ferreira CS, Silva CP, Ribeiro AP, de Castro Moreira Santos Júnior A, de Paula Brandão PR, Silva APB, da Rocha MCO, Xavier ME, Titze-de-Almeida SS, Shimizu HE, Delgado-Rodrigues RN. Persistent, new-onset symptoms and mental health complaints in Long COVID in a Brazilian cohort of non-hospitalized patients. BMC Infect Dis. 2022 Feb 8;22(1):133. doi: 10.1186/s12879-022-07065-3.

8.2) Kubota T, Kuroda N, Sone D. Neuropsychiatric aspects of long COVID: A comprehensive review. Psychiatry Clin Neurosci. 2023 Feb;77(2):84-93. doi: 10.1111/pcn.13508.

8.3) Efstathiou V, Stefanou MI, Demetriou M, Siafakas N, Makris M, Tsivgoulis G, Zoumpourlis V, Kympouropoulos SP, Tsoporis JN, Spandidos DA, Smyrnis N, Rizos E. Long COVID and neuropsychiatric manifestations (Review). Exp Ther Med. 2022 May;23(5):363. doi: 10.3892/etm.2022.11290.

8.4) Davis, H.E., McCorkell, L., Vogel, J.M. et al. Long COVID: major findings, mechanisms and recommendations. Nat Rev Microbiol 21, 133–146 (2023). https://doi.org/10.1038/s41579-022-00846-2.

8.5) Asadi-Pooya AA, Nemati M, Nemati H. 'Long COVID': Symptom persistence in children hospitalised for COVID-19. J Paediatr Child Health. 2022 Oct;58(10):1836-1840. doi: 10.1111/jpc.16120.

Experimental design

The authors should address the small sample size issue, as the study included only 62 cases and 52 controls. Increasing the sample size would strengthen the generalizability of the findings.

Validity of the findings

The authors should clarify the specific duration of Long COVID (5-8 months) in relation to the diagnosis, providing a clearer understanding of the study population.

The authors should add additional participant information, such as demographics and relevant comorbidities, should be included to provide a more comprehensive understanding of the study population.

The authors should explicitly highlight the limitations of the study, such as the small sample size and potential selection bias, to address any factors that may impact the interpretation of the findings.

How did the authors precisely evaluate post-COVID sadness and anxiety? It is essential to consider that individuals can experience sadness or anxiety for numerous reasons, including financial difficulties, relationship issues, work conflicts, unemployment, or even waking up in a low mood. How can we effectively distinguish these factors from the specific effects attributed to Long COVID? This aspect appears to be a significant limitation within the article.

---

## Round 0.2 · Major Revisions

Your manuscript has undergone review by two experts in the field. Their feedback, as outlined below, acknowledges the significance of your work while also highlighting specific areas that require further attention. Specifically, concerns were expressed regarding the experimental design and the statistical results. We are interested in the possibility of publishing your study, contingent upon your response to these concerns through a revised manuscript. Therefore, we invite you to carefully address the raised points and resubmit your manuscript

Reviewer 1 ·

Basic reporting

The addition of new tables now provides an adequate description. There are also details now regarding the population etc. The language has also been improved, and additional text now attempts to justify the authors' design.

Experimental design

This area remains unclear to me.
1. If the previous work showed an association (in the same sample) then there is little novelty to the new finding - the same association has merely been explored with a new design. it is unclear what this study adds.

2. if the hypothesis is of an association duE to "shared brain pathology", then again the use of two out of three associated features as a classifier and then studying the third as an outcome is difficult to justify.

Validity of the findings

the statistics presented in table 1 are incorrect. it appears as if the authors have run separate Fisher's tests by group for the two levels of the predictor variable (as separate p values are reported).

Reviewer 2 ·

Basic reporting

NA

Experimental design

NA

Validity of the findings

NA

Additional comments

The authors diligently addressed all questions posed by both reviewers, providing thorough point-by-point responses. Furthermore, they incorporated the requisite revisions into the updated manuscript, rendering it suitable for acceptance at PeerJ.

---

## Round 0.3 · accepted · Accept

Thank you for submitting the revised manuscript and response letter. I am delighted to inform you that your manuscript titled "Sleep and Memory Complaints in Long COVID: An Insight into Clustered Psychological Phenotypes" has been accepted for publication in PeerJ.